# Snijders Blok–Campeau Syndrome: Description of 20 Additional Individuals with Variants in *CHD3* and Literature Review

**DOI:** 10.3390/genes14091664

**Published:** 2023-08-23

**Authors:** Patricia Pascual, Jair Tenorio-Castano, Cyril Mignot, Alexandra Afenjar, Pedro Arias, Natalia Gallego-Zazo, Alejandro Parra, Lucia Miranda, Mario Cazalla, Cristina Silván, Delphine Heron, Boris Keren, Ioana Popa, María Palomares, Emi Rikeros, Feliciano J. Ramos, Berta Almoguera, Carmen Ayuso, Saoud Tahsin Swafiri, Ana Isabel Sánchez Barbero, Varunvenkat M. Srinivasan, Vykuntaraju K. Gowda, Manuela Morleo, Vicenzo Nigro, Stefano D’Arrigo, Claudia Ciaccio, Carmen Martin Mesa, Beatriz Paumard, Gema Guillen, Ana Teresa Serrano Anton, Marta Domínguez Jimenez, Veronica Seidel, Julia Suárez, Valerie Cormier-Daire, The SOGRI Consortium, Julián Nevado, Pablo Lapunzina

**Affiliations:** 1CIBERER, Center for Biomedical Research in Rare Diseases Network, 28029 Madrid, Spain; patripascual0@gmail.com (P.P.); jairantonio.tenorio@gmail.com (J.T.-C.); palajara@gmail.com (P.A.); nataliagallegozazo@gmail.com (N.G.-Z.); alejandro.parra.ingemm@gmail.com (A.P.); luciamiranda.ingemm@gmail.com (L.M.); mpalomares.ingemm@gmail.com (M.P.); emikarina.rikeros@salud.madrid.org (E.R.); framos@unizar.es (F.J.R.); balmoguera@quironsalud.es (B.A.); cayuso@fjd.es (C.A.); stahsin@quironsalud.es (S.T.S.); ana.sbarbero@quironsalud.es (A.I.S.B.); thesogriconsortium@gmail.com (The SOGRI Consortium); jnevadobl@gmail.com (J.N.); 2INGEMM-IdiPaz, Institute of Medical and Molecular Genetics, 28046 Madrid, Spain; mario.cazalla16@gmail.com (M.C.); cristina_sf8@hotmail.com (C.S.); 3ITHACA, European Reference Network, 1140 Brussels, Belgium; stefano.darrigo@istituto-besta.it (S.D.); claudia.ciaccio@istituto-besta.it (C.C.); 4Département de Génétique, APHP Sorbonne Université, 75013 Paris, France; delphine.heron@aphp.fr (D.H.); cyril.mignot@aphp.fr (C.M.); alexandra.afenjar@aphp.fr (A.A.); boris.keren@aphp.fr (B.K.); ioana.popa@aphp.fr (I.P.); 5Centre de Réference Déficiences Intellectuelles de Causes Rares, 75013 Paris, France; 6Unidad de Genética Clínica, Servicio de Pediatría, Hospital Clínico Universitario ‘Lozano Blesa’, Facultad de Medicina, Universidad de Zaragoza, IIS-Aragón Grupo B32-20R, 50013 Zaragoza, Spain; 7Department of Genetics and Genomics, Fundación Jiménez Díaz University Hospital, Health Research Institute Fundación Jiménez Díaz (IIS-FJD), 28040 Madrid, Spain; 8Department of Pediatric Neurology, Indira Gandhi Institute of Child Health, Bangalore 560029, India; varunms951@gmail.com (V.M.S.); drknvraju@hotmail.com (V.K.G.); 9Telethon Institute of Genetics and Medicine (TIGEM), 80078 Pozzuoli, Italy; morleo@tigem.it (M.M.); nigro@tigem.it (V.N.); 10Department of Precision Medicine, University of Campania “Luigi Vanvitelli”, 80138 Naples, Italy; 11Department of Pediatric Neurosciences, Fondazione IRCCS Istituto Neurologico Carlo Besta, 20126 Milan, Italy; 12HM Hospitales, 28660 Madrid, Spain; cmartin@genologica.com (C.M.M.); beatriz.paumard@gmail.com (B.P.); gemguil@gmail.com (G.G.); 13Department of Medical Genetics, Hospital Clínico Universitario Virgen de la Arrixaca, IMIB-Arrixaca, 30120 Murcia, Spain; anateserrano@gmail.com (A.T.S.A.); aylenmarchis@gmail.com (M.D.J.); 14Genomics Unit, HGU Gregorio Marañón, 28007 Madrid, Spain; veronicaadriana.seidel@salud.madrid.org (V.S.); julia.suarez@salud.madrid.org (J.S.); 15Department of Genomic Medicine for Rare Diseases, INSERM UMR1163, Imagine Institute, Necker Enfants Malades Hospital, Paris Cité University, 75015 Paris, France; valerie.cormier-daire@inserm.fr

**Keywords:** *CHD3*, Snijders Blok–Campeau syndrome, overgrowth, neurodevelopmental disorders, SNIBCPS

## Abstract

Snijders Blok–Campeau syndrome (SNIBCPS, OMIM# 618205) is an extremely infrequent disease with only approximately 60 cases reported so far. SNIBCPS belongs to the group of neurodevelopmental disorders (NDDs). Clinical features of patients with SNIBCPS include global developmental delay, intellectual disability, speech and language difficulties and behavioral disorders like autism spectrum disorder. In addition, patients with SNIBCPS exhibit typical dysmorphic features including macrocephaly, hypertelorism, sparse eyebrows, broad forehead, prominent nose and pointed chin. The severity of the neurological effects as well as the presence of other features is variable among subjects. SNIBCPS is caused likely by pathogenic and pathogenic variants in *CHD3 (Chromodomain Helicase DNA Binding Protein 3)*, which seems to be involved in chromatin remodeling by deacetylating histones. Here, we report 20 additional patients with clinical features compatible with SNIBCPS from 17 unrelated families with confirmed likely pathogenic/pathogenic variants in *CHD3*. Patients were analyzed by whole exome sequencing and segregation studies were performed by Sanger sequencing. Patients in this study showed different pathogenic variants affecting several functional domains of the protein. Additionally, none of the variants described here were reported in control population databases, and most computational predictors suggest that they are deleterious. The most common clinical features of the whole cohort of patients are global developmental delay (98%) and speech disorder/delay (92%). Other frequent features (51–74%) include intellectual disability, hypotonia, hypertelorism, abnormality of vision, macrocephaly and prominent forehead, among others. This study expands the number of individuals with confirmed SNIBCPS due to pathogenic or likely pathogenic variants in *CHD3.* Furthermore, we add evidence of the importance of the application of massive parallel sequencing for NDD patients for whom the clinical diagnosis might be challenging and where deep phenotyping is extremely useful to accurately manage and follow up the patients.

## 1. Introduction

Snijders Blok–Campeau syndrome (SNIBCPS) (MIM #618205) is a notably infrequent autosomal dominant disease, first described by Snijders Blok et al. in 2018 [1] and caused by pathogenic and likely pathogenic variants in the Chromodomain Helicase DNA Binding Protein 3 (*CHD3*) gene. The disorder shows an autosomal dominant pattern of inheritance. The most common clinical findings in affected patients include a neurodevelopmental disorder, characterized by global or speech delay, followed by either intellectual disability (ID) or specific learning difficulties, behavioral disorders like autism spectrum disorder (ASD) or attention deficit hyperactivity disorder (ADHD), and dysmorphic features including macrocephaly, prominent forehead, hypertelorism, face asymmetry, deeply set eyes, strabismus, epicanthus, wide nasal bridge, prominent nose, pointed chin, high palate and joint laxity among others [1]. The clinical features are highly heterogeneous and variable among individuals. Initially, 35 patients were described carrying a de novo pathogenic variant that disrupted the *CHD3* gene [1]. To date, 63 patients have been diagnosed through molecular genetic techniques [1,2,3,4,5,6]. These patients had missense, in-frame deletions, nonsense and frameshift pathogenic or likely pathogenic variants. Additionally, one single case had a complete deletion of the *CHD3* gene, and one patient was found to have a large duplication of the *CHD3* gene [2].

Mammalian chromodomains helicase DNA-binding (CHDs) comprises a large family of proteins that have an indispensable role in developmental processes. Specifically, these proteins are involved in transcriptional regulation, and they can exert their chromatin remodeling activity to form the core ATPase subunit of the complex Nucleosome Remodeling Deacetylase (NuRD), which is associated with multiple cellular processes such as genomic integrity, cell cycle progression, or embryonic stem cell differentiation. Specifically, *CHD3* encodes for a protein involved in late neural radial migration and cortical layer specification [2]. All these proteins have two chromodomains and two helicase domains. However, CHD3 additionally has two plant homeodomains (PHD), the function of which is to be responsible for the physical binding of CHDs to the NuRD chromatin-remodeling complex. Finally, it also contains domains of unknown function (DUF) [7] (Figure 1).

In addition, according to previous findings and those described in this report, a variable degree of expression and reduced penetrance of the disease has been suggested [8]. Recently, it was observed that the majority of the variants that were inherited were maternally inherited. As mentioned by Spek et al. [8], the only parent in their cohort from whom a variant was inherited was affected, suggesting that female gender protects against genetic variation in disease.

Here, we describe 20 additional patients from 17 unrelated families with confirmed novel pathogenic variants in *CHD3* and review the clinical and molecular characteristics of a total of 63 patients from the literature. Our work provides an updated clinical phenotype of patients with SNIBCPS and discusses the molecular pathogenesis and mechanism of inheritance. It confirms intrafamilial phenotypic variability. This article has special relevance for all professionals who are in charge of follow up patients with neurodevelopmental disorders (NDDs) and highlights the importance of an early diagnosis and the implementation of massive parallel sequencing technologies such as whole exome sequencing as the first tier for the diagnosis of these highly heterogeneous disorders.

## 2. Materials and Methods

### 2.1. Patient Cohort

All patients were recruited based upon either their clinical findings or the molecular defect in *CHD3* from laboratories and hospitals in Spain, India, Italy and France. Informed consent was obtained from all patients and/or their legal guardians. Neuropsychiatric findings were evaluated by a specialist for each patient. Some of the patients from Hospital Universitario La Paz are part of the Spanish Overgrowth Registry Initiative (SOGRI) database.

### 2.2. Genetic Analysis

Patients were mostly analyzed by whole exome sequencing (WES). Segregation analysis was performed in parental samples by Sanger sequencing when samples were available. Variants are described using the HGVS nomenclature and classified according to the ACMG guidelines [9]. The in silico pathogenicity of each of the variants was assessed using CADD-PHRED v1.6 and REVEL v4.3 scores.

## 3. Results

By means of the application of massive parallel sequencing technologies, as well as international collaboration through scientific networks, we have identified 20 additional cases with SNIBCPS with pathogenic or likely pathogenic variants in *CHD3*. Segregation analysis showed that variants were de novo in 16 out of 20 patients (80%), inherited in 3 others (15%), and segregation analysis was not possible for one of them.

At the clinical level, all 20 patients were phenotyped, and the identified clinical features are summarized in Table 1. The most frequent (>75%) clinical features observed in these patients included global developmental delay and speech disorder/delay. Although global developmental delay was almost constant, including generalized hypotonia and language delay, almost half of the patients aged 5 years old and beyond showed no ID. Two of our patients presented with ASD, and six of them with ADHD.

In addition, in our cohort we present a family (P12, 13 and 14, Table 2) in which mother and two children have variants in the *CHD3* gene. Both mother and daughter presented with initial global developmental delay but had no ID and a diagnosis of developmental coordination disorder. The son presented with ASD and moderate ID.

Other frequent clinical features (50–75%) observed were: intellectual disability, hypotonia, hypertelorism, abnormality of vision, macrocephaly and prominent forehead.

At the molecular level, 16 out the 20 variants detected were missense and the other four included a nonsense, a frameshift deletion, and two in-frame deletions. All variants were heterozygous.

The majority of the detected variants were located within or between the helicase ATP-binding domain, the helicase C-terminal domain and the DUF domain except for the frameshift variant, which is located before the C-terminal 2 domain and the transcript encoded is predicted to be degraded by the nonsense-mediated decay (NMD) mechanism (Figure 1).

After ACMG variant classification, eight variants were classified as pathogenic and 12 as likely pathogenic. None of the variants reported herein were previously reported and/or associated with SNIBCPS. Moreover, none of the variants described in our series have previously been reported in pseudo-control population databases (gnomAD exomes, gnomAD genomes, Kaviar, Beacon, 1000G, ESP and Bravo) and the majority of the computational evidence support a deleterious/pathogenic effect for all the variants reported in this study (Table 2).

## 4. Discussion

NDDs comprise a large and highly heterogeneous group of disorders, caused by a variety of pathogenic and likely pathogenic variants in >1000 genes and in which there might by a highly clinical presentation overlap. With the advances in massive parallel sequencing technologies, it has become possible to identify and confirm or discard initial clinical suspicions in this group of patients.

SNIBCPS was first described in 2018 by Snijders Blok et al. [1]. SNIBCPS can be classified as an NDD, and because it is caused by pathogenic variants in *CHD3*, it may have phenotypic overlap with other human diseases caused by mutations in CHD proteins. Some of these may include *CHD7* causing CHARGE syndrome (MIM #214800), *CHD4* as causative of Sifrim–Hitz–Weiss syndrome (MIM #603277), or *CHD8* causing autism spectrum disorder (MIM #610528).

One of the challenges in the recognition of the SNIBCPS syndrome is the highly phenotypic overlap with other genetic disorders, especially NDDs and overgrowth syndromes, because the most common findings are macrocephaly and neurodevelopmental problems. Besides NDDs and enlarged head circumference, some dysmorphic features such as pointed chin or frontal bossing may resemble syndromes such as Sotos, Malan, Tenorio, or Simpson–Golabi–Behmel syndromes [10,11,12,13].

NuRD is an ATP-dependent chromatin remodeling complex. It consists of multiple proteins, including *CHD3* and *CHD4*, which are important for the activity of the NuRD complex, as they provide the energy and helicase function necessary for chromatin remodeling. This complex has important functions in cellular processes such as peripheral nerve myelination and cortical development in the brain. CHD3 is an ATP-dependent chromatin remodeling protein that serves as core member of the NuRD complex [14] and it has been shown to play an important role in the viability of the developing embryonic brain [15].

In this study, we report 20 patients from 17 unrelated families with a clinical phenotype compatible with SNIBCPS in whom a pathogenic/likely pathogenic variant in the *CHD3* gene was identified. Thus, thanks to the increase in the number of patients with *CHD3* defects, it was possible to expand the SNIBCPS phenotype.

Reviewing our cohort of patients, we observed that global developmental delay and speech disorder/delay are the most common (>75%) clinical features. The clinical characteristics observed in 50–75% of individuals were intellectual disability, hypotonia, hypertelorism, abnormality of vision, macrocephaly and prominent forehead. Less common clinical features (25–50%) were abnormality of CNS, autism and some dysmorphic features such as pointed chin or wide nasal bridge, among others (Table 1). In addition, we were able to expand the phenotypic spectrum with other unreported features such as epicanthus (12%), attention deficit hyperactivity disorder (ADHD) (7%), high palate (6%), foot deformities (5%), depressed nasal root (5%) and blushed cheeks (4%). Our series confirms intrafamilial clinical variability highlighting the importance of paying attention to the variants transmitted when interpreting WES or WGS analyses.

We found 20 likely pathogenic/pathogenic variants. On the other hand, it has previously been reported that most of the variants described are located within the ATPase/helicase domain of the *CHD3* gene [2].

Of the 20 individuals in our cohort, eight of them had pathogenic variants in the ATPase/helicase domain, with 12 variants being outside this domain. It has previously been reported that there are no phenotypic differences between patients with variants within the ATPase/helicase domain and individuals with variants outside this domain [2].

In our series, we also found no phenotypic differences between patients with a variant within the ATPase/helicase domain and those with a variant outside of this domain.

*CHD3* is extremely intolerant to LoF (loss-of-function) and missense variation based on in silico analysis (probability of LoF intolerance = 1, observed/expected = 0.09 [0.05–0.15]; Z-score = 6.15, observed/expected = 0.5 [0.46–0.53]). These values suggest that a haploinsufficiency mechanism may be the cause of the disease, as other studies have already suggested [8]. We analyzed the distribution of CCRs (Constrained Coding Regions) and found that four of the variants present in our cohort of patients had highly constrained regions with percentiles in the interval (90–100).

In addition, previous studies have sought to determine the effect of variants in *CHD3* using an ATPase assay. A clear decrease in ATPase activity was observed in two variants (p.Arg1121Pro and p.Arg1172Gln) [1]. As mentioned above, in our study, 13 variants were found in the ATPase functional domain, half of them in the amino acid arginine (Figure 1). Moreover, it was observed that ATP binds strongly to Arg and with high affinity, and Arg dominates the direct binding of ATP and affects the self-association of accumulated ATPs. The size of the ATP pool is effectively regulated by the distribution of Arg [16]. Our hypothesis regarding these findings is that the effect of these variants on *CHD3* (p.Arg985Trp, p.Arg1044Trp, p.Arg1070Gln and p.Arg1169Trp) may affect critical functions of the protein, such as impacting the ability of the protein to interact with other molecules or proteins involved in chromatin remodeling processes. On the other hand, it can influence the stability and folding capacity of the protein to adopt its three-dimensional structure, leading to protein misfolding and/or instability. In order to fully understand the consequences of the different variants in *CHD3*, further functional analysis and protein modeling techniques are needed.

In our series, segregation analysis showed that three of the patients had a variant inherited from one of their parents (15%). In one of these families, the transmitting parent was apparently healthy. It has been reported that most of the variants inherited in other cohorts were maternally inherited, suggesting that female gender protects against genetic variations in this disease. This observation may indicate incomplete penetrance and variable expressivity in this syndrome [8].

In conclusion, we herein report 20 additional patients with novel variants along the *CHD3* gene. The main clinical manifestations are the presence of neurodevelopmental problems without intellectual disability in half of the patients as well as dysmorphic features, such as macrocephaly. At the molecular level, we found in our cohort, as well as in the cohorts described above, that most of the variants are located in the ATPase/helicase functional domain. Thus, we have increased the number of patients with SNIBCPS as well as expanded the phenotypic features of this unusual disorder. Further studies may help to elucidate genotype–phenotype correlations and to understand the mechanism by which a *CHD3* gene defect causes Snijders Blok–Campeau syndrome (OMIM #518205).

## Figures and Tables

**Figure 1 genes-14-01664-f001:**
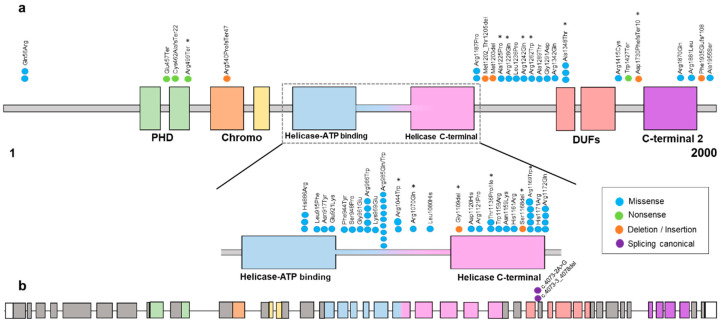
(**a**) Schematic representation of the variants found in the CHD3 protein (transcript 1, NM_001005273.3) except for the splicing canonical variant that is shown in (**b**), found in our cohort and in the literature [1,2,3,4,5,6]. Most of the variants are found within the ATPase/helicase functional domain. The colors of missense, nonsense, deletion/insertion and splicing canonical are shown in the legend. * Variants reported in this study. (**b**) Schematic representation of the *CHD3* exons (transcript 1, NM_001005273.3) with the splicing canonical variants.

**Table 1 genes-14-01664-t001:** Clinical features found in patients with SNIBCPS. HPO prioritization frequency of clinical features described in patients with *CHD3* variants. Percentage was calculated according to the total number of patients reported in the literature. Patients compiled from [1,2,3,4,5,6] (63 patients) and the 20 patients reported herein.

Frequency	Snijders Blok–Campeau Patients	Drivas et al., 2020	Mizukami et al., 2021	Xi-Yong, 2021	Coursimault et al., 2021	Snijders Blok et al., 2018	LeBreton et al., 2022	This Study (n = 20)	Total (n = 83)	Percentage (%)
	HP:0001263 Global developmental delay	24		1	1	35	1	19	81	98%
>75%	HP:0002167 Speech disorder/delay	24	1	1	1	33	1	15	76	92%
	HP:0001249 Intellectual disability	20	1	1	1	27		11	61	73%
	HP:0001252 Hypotonia	22	1	1	1	21		13	59	71%
	HP:0000316 Hypertelorism	13	1	1		24		13	52	63%
51–74%	HP:0000504 Abnormality of vision	18		1	1	23		3	46	55%
	HP:0000256 Macrocephaly	10		1	1	19		15	46	55%
	HP:0000244 Prominent forehead				1	28		14	43	52%
	HP:0002011 Abnormality of CNS	9			1	16		9	35	42%
	HP:0002007 Frontal bossing	13				11	1	10	35	42%
	HP:0000307 Pointed chin	12			1		1	13	27	33%
	HP:0000431 Wide nasal bridge	17		1				8	26	31%
26–50%	HP:0000219 Thin upper lips	17						9	26	31%
	HP:0001388 Joint laxity		1	1	1	12		10	25	30%
	HP:0000717 Autism spectrum disorder	9	1		1	9	1	2	23	28%
	HP:0000486 Strabismus	6	1	1	1	10		4	23	28%
	HP:0000293 Full cheeks	13					1	8	22	27%
	HP:0005338 Sparse lateral eyebrow	12	1		1			5	19	23%
	HP:0001270 Motor delay			1	1			17	19	23%
	HP:0000369 Low-set ears	8	1	1	1		1	5	17	20%
	HP:0000490 Deep-set eyes	13					1	2	16	19%
	HP:0031936 Delayed walking							16	16	19%
	HP:0045025 Narrow palpebral fissure	10	1	1				3	15	18%
	HP:0000506 Telecanthus	10			1		1	1	13	16%
	HP:0000448 Prominent nose	6	1		1			4	12	14%
	HP:0012371 Mid-face hypoplasia	9			1			2	12	14%
	HP:0001388 Abnormality of the male genitalia				1	6		5	12	14%
	HP:0001627 Congenital heart defects	5				3		3	11	13%
	HP:0001250 Seizures	5				4		2	11	13%
	HP:0008872 Feeding difficulties in infancy					10		1	11	13%
	HP:0000358 Posteriorly rotated ears	9					1		10	12%
	HP:0000337 Broad forehead							10	10	12%
	HP:0000286 Epicanthus							10	10	12%
	HP:0006349 Absent teeth	5				2		1	8	10%
	HP:0007018 Attention deficit hyperactivity disorder							1	7	8%
	HP:0100790 Hernia					5		6	6	7%
<25%	HP:0000455 Broad nasal tip	6						1	6	7%
	HP:0000164 Dental abnormalities				1			5	6	7%
	HP:0000733 Stereotypic behavior				1			4	5	6%
	HP:0000218 High palate							5	5	6%
	HP:0001760 Foot deformities							4	4	5%
	HP:0002119 Ventriculomegaly				1			3	4	5%
	HP:0005280 Depressed nasal root							4	4	5%
	HP:0001041 Blushed cheeks							3	3	4%
	HP:0000365 Hearing impairment	3							3	4%
	HP:0001260 Dysarthria							3	3	4%
	HP:0000252 Microcephaly	2				1			3	4%
	HP:0002317 Unsteady gait							2	2	2%
	HP:0000957 Cafe-au-lait spot /HP:0007565 Multiple cafe-au-lait spots							2	2	2%
	HP:0045025 Narrow palpebral fissure	10						3	2	2%
	HP:0040082 Happy demeanor				1			1	2	2%
	HP:0000753 Autism with high cognitive abilities							1	1	1%
	HP:0000077 Abnormality of the kidney				1				1	1%
	HP:0000664 Synophris							1	1	1%
	HP:0000322 Short philtrum							1	1	1%
	HP:0000160 Narrow mouth							1	1	1%
	HP:0004442 Sagittal craniosynostosis							1	1	1%
	HP:0001047 Pes planus							1	1	1%
	HP:0001047 Atopic dermatitis							1	1	1%
	HP:0045074 Thin eyebrow							1	1	1%

**Table 2 genes-14-01664-t002:** Variants detected in *CHD3* in our cohort. ACMG, American College of Medical Genetics. ^#^ Population frequency was estimated from pseudo-control databases: GnomAD genomes (v3.0); GnomAD exomes (v3.1); Kaviar (version160204-Public); Beacon (v2.0); 1000 G; Phase III; and Bravo (TOVMed Freeze 8).

Family	Proband	Genomic Coordinate (hg38)	cDNA and Protein Location	Exon/Intron	Mutation Type	Zygosity	Inheritance	Population Frequency #	CADD	REVEL	ACMG Prediction
1	1	chr17:7811001	NM_001005271.3:c.5184_5185del (p.Asp1730PhefsTer10)	33	Frameshift	Heterozygous	De novo	-	35	-	Likely Pathogenic
2	2	chr17:7804024	NM_001005271.3:c.3130C>T (p.Arg1044Trp)	18	Missense	Heterozygous	De novo	-	29.8	-	Pathogenic
3	3	chr17:7804650	NM_001005273.3:c.3209G>A (p.Arg1070Gln)	20	Missense	Heterozygous	De novo	0.0000159	24.9	0.161	Likely Pathogenic
4	4	chr17:7806590	NM_001005271.3:c.3673G>C (p.Ala1225Pro)	22	Missense	Heterozygous	De novo	-	33	-	Pathogenic
5	5	chr17:7805996	NM_001005271.3:c.3325_3327delGGT (p.Tyr1109del)	20	In frame deletion	Heterozygous	De novo	-	-	-	Likely Pathogenic
6	6	chr17:7806642	NM_001005271.3:c.3725G>A (p.Arg1242Gln)	23	Missense	Heterozygous	Inherited from father	-	32	0.947	Likely Pathogenic
7	7	chr17:7806600	NM_001005271.3:c.3683G>A (p.Arg1228Gln)	23	Missense	Heterozygous	De novo	-	31	0.957	Pathogenic
8	8	chr17:7903278	NM_001005273.3:c.3502_3504del (p.Ser1168del)	23	In frame deletion	Heterozygous	De novo	-	-	-	Likely Pathogenic
9	9	chr17:7902972	NM_001005273.3:c.3406A>C (p.Thr1136Pro)	22	Missense	Heterozygous	De novo	-	29.9	0.966	Likely Pathogenic
10	10	chr17:7895142	NM_001005273.3:c.1495C>T (p.Arg499Ter)	9	Nonsense	Heterozygous	De novo	-	-	-	Pathogenic
11	11	chr17:7903962	NM_001005273.3:c.3865G>A (p.Ala1289Thr)	24	Missense	Heterozygous	Unknown	-	29.1	0.831	Likely Pathogenic
12	12	chr17:7903962	NM_001005271.3:c.4042G>A (p.Ala1348Thr)	24	Missense	Heterozygous	Inherited from mother	-	29.1	0.831	Likely Pathogenic
12	13	chr17:7903962	NM_001005271.3:c.4042G>A (p.Ala1348Thr)	24	Missense	Heterozygous	Inherited from mother	-	29.1	0.831	Likely Pathogenic
12	14	chr17:7903962	NM_001005271.3:c.4042G>A (p.Ala1348Thr)	24	Missense	Heterozygous	Inherited from mother	-	29.1	0.831	Likely Pathogenic
13	15	chr17:7900707	NM_001005273.3:c.2954G>A (p.Arg985Gln)	18	Missense	Heterozygous	De novo	-	32	0.777	Pathogenic
14	16	chr17:7903281	NM_001005273.3:c.3505C>T p.Arg1169Trp	23	Missense	Heterozygous	De novo	-	25.7	0.899	Pathogenic
15	17	chr17:7903317	NM_001005273.3:c.3541A>T p.Ile1181Phe	23	Missense	Heterozygous	De novo	-	28.4	0.904	Likely Pathogenic
15	18	chr17:7903317	NM_001005273.3:c.3541A>T p.Ile1181Phe	23	Missense	Heterozygous	De novo	-	28.4	0.904	Likely Pathogenic
16	19	chr17:7903282	NM_001005273.3:c.3506G>A p.Arg1169Gln	23	Missense	Heterozygous	De novo	-	31	0.96	Pathogenic
17	20	chr17:7804024	NM_001005271.3:c.3130C>T (p.Arg1044Trp)	18	Missense	Heterozygous	De novo	-	29.8	-	Pathogenic

## Data Availability

The data presented in this study are available in tables and figures of the article.

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
