# Peer review of "Snijders Blok–Campeau Syndrome: Description of 20 Additional Individuals with Variants in CHD3 and Literature Review"

_genes, 2023, doi:10.3390/genes14091664_

Round 1

Reviewer 1 Report

The authors presented an original study highlighting the detailed description of 19 new individuals diagnosed with Snijders-Blok-Campeau syndrome, presenting new genetic variants as well as showing the phenotypic heterogeneity related to the different CHD3 variants. Deep phenotyping associated with massive paralleled sequencing enabled the proper recognition of patients with complex phenotypes related to neurodevelopmental disorders. A minor suggestion for authors is to include, if possible, the description of neuroimaging features from the available individuals. 

Author Response

In addition, we would like to mention that during this time we have collected one more patient with defects in the CHD3 gene, making a total of 20 patients. 

Reviewer 2 Report

In this manuscript, the authors have reported 19 new cases of Snijders Blok-Campeau syndrome (SNIBCPS) and compiled all the cases ie new cases and reported cases to review the clinical and molecular characteristics of these patients. Considering SNIBCPS was only first described in 2018 with only 19 reported cases, any additional knowledge on the genotype and phenotype of these patients are critical for the diagnosis, understanding and treatment of SNIBCPS.

Overall the manuscripts was very well written, with an introduction that provided sufficient background and the cases analysed with great care. Just a few questions/suggestion

1) There appeared to be two different font size in the Abstract. The font size in line 47 to line 56 is smaller compared to the rest of the section.

2) Any information on the gender of the new cases

3) Considering that the majority of the cases are maternal inherited, a) what is the percentage of cases for heritance from either parents b) are there any common feature for the cases that are NOT maternal inherited

4) Also how can the authors explained new cases appeared not to be inherited/sporadic? Have the parents been examined?

5) For the new cases, are some of them family members and if so what are their relationship ie siblings etc

6) While the information presented in the table 1 is very detailed, it would be easier to understand if some of the more major presentations are presented as a graph
